# Low Bacterial Biomass in Human Pancreatic Cancer and Adjacent Normal Tissue

**DOI:** 10.3390/ijms26010140

**Published:** 2024-12-27

**Authors:** Michael S. May, Heekuk Park, Dalia H. Moallem, Dwayne Seeram, Sun Dajiang, Hanina Hibshoosh, Jacob K. Jamison, Anne-Catrin Uhlemann, Gulam A. Manji

**Affiliations:** 1Department of Medicine, Vagelos College of Physicians & Surgeons, Columbia University, 630 West 168th Street, New York, NY 10032, USA; hp2523@cumc.columbia.edu (H.P.); dalia.moallem@gmail.com (D.H.M.); ds4057@cumc.columbia.edu (D.S.); jaj7021@med.cornell.edu (J.K.J.); au2110@cumc.columbia.edu (A.-C.U.); gam2140@cumc.columbia.edu (G.A.M.); 2Department of Pathology, Vagelos College of Physicians & Surgeons, Columbia University, 630 West 168th Street, New York, NY 10032, USA; ds2429@cumc.columbia.edu (S.D.); hhh1@cumc.columbia.edu (H.H.); 3The Herbert Irving Comprehensive Cancer Center, Vagelos College of Physicians & Surgeons, Columbia University, 630 West 168th Street, New York, NY 10032, USA

**Keywords:** pancreatic cancer, microbiome, chemotherapy, gemcitabine

## Abstract

The gut microbiome plays an important role in the carcinogenesis of luminal gastrointestinal malignancies and response to antineoplastic therapy. Preclinical studies have suggested a role of intratumoral gammaproteobacteria in mediating response to gemcitabine-based chemotherapy in pancreatic ductal adenocarcinoma (PDAC). To our knowledge, this is the first study to evaluate the impact of the PDAC microbiome on chemotherapy response using samples from human pancreatic tumor resections. We performed 16S rRNA gene amplification and sequencing on both formalin-fixed paraffin-embedded (FFPE) and fresh frozen human PDAC resection samples. We analyzed frozen samples from 26 patients with resected PDAC and examined tumor and tumor-adjacent normal tissue. These patients represented nine long-term survivors (LTS) and nine short-term survivors (STS) after neoadjuvant gemcitabine therapy and eight control patients who did not receive any neoadjuvant therapy prior to resection. We also included FFPE samples from five patients, including tumor samples (3 samples per patient), tumor-adjacent normal tissue (2 per patient) and tumor-adjacent paraffin (1 per patient). Within frozen tissue, total DNA yields were high, but bacterial DNA was generally low, comparable to those seen in negative controls. In FFPE tissue, DNA yields were low and bacterial abundances were comparable in paraffin, tumor and normal PDAC samples. Gammaproteobacteria concentrations did not correlate with outcomes in patients treated with neoadjuvant gemcitabine-based chemotherapy. Our study found low microbial biomass in pancreatic tumor tissue, with no detectable association between bacterial taxa and chemotherapy outcomes. These results suggest a limited role of the microbiome in gemcitabine-based chemotherapy response in PDAC. Preclinical studies have implicated the pancreatic tumor microbiome in driving response to therapy. Cytidine deaminase, an enzyme produced by gammaproteobacteria, can metabolize gemcitabine and has been hypothesized to inhibit pancreatic tumor response to chemotherapy. Several clinical trials have evaluated the role of the tumor microbiome in pancreatic cancer treatment. We evaluated the impact of the pancreatic tumor microbiome on chemotherapy response using samples from human pancreatic tumor resections. We found a low microbial load that is partially attributable to contaminants and that gammaproteobacteria levels did not correlate with outcomes in patients with pancreatic cancer treated with gemcitabine-based chemotherapy.

## 1. Introduction

The role of the microbiome in carcinogenesis and response to cancer treatment is a topic of increasing interest. The gut microbiome has high microbial abundance at multiple sites and has been strongly implicated in carcinogenesis [1,2,3] and response to immunotherapy [4]. The abundance and oncologic impact of the microbiome at nonluminal sites is the subject of ongoing exploration and debate [5].

The pancreas lives between these two worlds: directly connected to the duodenum and in close proximity to luminal gut microbes, but not part of the gut lumen. Preclinical studies of pancreatic cancer have implicated a role of the tumor microbiome in pancreatic cancer carcinogenesis and response to therapy [6,7,8]. Geller et al. noted that the long isoform of cytidine deaminase, an enzyme produced by gammaproteobacteria, was capable of metabolizing gemcitabine into its inactive form. In a colon cancer mouse model, the investigators were able to induce gemcitabine resistance through the introduction of intratumoral gammaproteobacteria, and antibiotic therapy was able to reverse this effect. They were then able to identify gammaproteobacteria within the tumor of 86/113 human pancreatic tumors [6].

However, interpreting these findings has been challenging due to the inherent limitations of microbiome studies, including sample contamination during surgical extraction, storage-related DNA degradation, and false-positive microbial detection from environmental sources. Previous clinical studies in human pancreatic ductal adenocarcinoma (PDAC) have produced conflicting results regarding the abundance and oncologic significance of the tumor microbiome, adjacent pancreatic tissue, and precancerous lesions [7,9,10,11]. This study aims to address these limitations by applying stringent contamination controls, including negative controls, bioinformatic decontamination using SCRuB, and comprehensive statistical analyses to evaluate potential associations between microbial populations and chemotherapy response in PDAC.

We sought to evaluate the microbial composition within primary tumors and adjacent pancreatic tissue, using a large biorepository, and correlate these findings with treatment and clinical outcomes. We hypothesized that increased levels of gammaproteobacteria in the intratumoral microbiome would correlate with inferior response to therapy and survival in patients treated with neoadjuvant gemcitabine-containing chemotherapies.

## 2. Results

### 2.1. Patient Characteristics

Of the five patients whose FFPE samples were analyzed, the median age was 67. Three of these patients received neoadjuvant gemcitabine and nab-paclitaxel (Table 1). There were 25 patients whose frozen samples were analyzed (nine long-term survivors [LTS], eight short-term survivors [STS] and eight untreated). The median age in the three groups was similar (67–71 years) and all cohorts included both head of pancreas and body/tail of pancreas tumors (Table 1). Median overall survival was 6.6 years in the LTS cohort, 1.4 years in the STS cohort and 4.0 years in the cohort that did not receive neoadjuvant chemotherapy. The median recurrence-free survival was 6.6 years in the LTS cohort, 1.1 years in the STS cohort and 1.3 years in the untreated cohort.

### 2.2. qPCR Results for FFPE and Fresh Frozen Samples with 16S V3V4 and V4 Amplification

DNA yields from FFPE samples varied significantly, ranging from 0.1 ng/µL to 9.8 ng/µL. Some samples had no detectable DNA, and many failed to amplify using qPCR with V3-V4 16S rRNA primers, suggesting challenges with this region (Table 2). In contrast, bacterial qPCR Ct values for the V4 region ranged from 31.6 to 41.8, though two tumor samples from a single patient (P4T1 and P4T3) did not show Ct values for this region.

Fresh frozen samples displayed a wide range of DNA yields, from 0.1 ng/µL to 98 ng/µL (Table 3). Two samples (SN2 and ST4) lacked V3V4 Ct values, possibly indicating difficulties in amplifying this region in those samples. Ct values for the V3V4 region ranged from 17.1 to 41.4, while the V4 region showed more consistent values between 16.4 and 42.1. Therefore, the V4 region was chosen for downstream analysis due to its more reliable and consistent amplification results.

### 2.3. 16S Microbiome FFPE Samples

Out of the initial 30 FFPE samples subjected to 16S microbiome analysis, 29 were found suitable for further downstream analysis. One of the normal samples for the fifth patient (P5N1) was excluded due to the presence of mitochondrial sequence reads. The 29 analyzed samples comprised normal tissues (*n* = 9), tumor tissues (*n* = 15), and paraffin-embedded samples (*n* = 5).

Figure 1 provides a visual representation of the taxonomy abundance for each group. At the phylum level, the FFPE samples predominantly exhibited the presence of Actinobacteriota, Firmicutes, Proteobacteria, Acidobacteriota and Bacteroidota.

Bacterial 16S qPCR yields were low regardless of tissue type (tumor, normal or paraffin), reflected by their similarity in Ct count to negative control (Table 2). Diversity analyses were also conducted to understand the microbial diversity within and between the samples. The Shannon index and Chao1 were employed to assess alpha-diversity, reflecting microbial richness and evenness within tissue types. These indices were chosen due to their widespread use in microbial ecology for detecting diversity patterns. No significant differences were observed across groups (Appendix A; Kruskal–Wallis tests, *p* = 0.5708 for Chao1 and *p* = 0.8136 for the Shannon index), indicating comparable microbial diversity between tumor and normal tissue samples. To evaluate beta-diversity, we conducted principal coordinates analysis (PCoA) based on Bray–Curtis dissimilarity distances, which visualizes differences in microbial community composition. The PCoA plots revealed no significant clustering among the samples, suggesting minimal differentiation in microbial composition across tissue types.

Differential abundance tests were conducted between normal and tumor tissue samples. In the normal tissue samples, *Staphylococcus saprophyticus* and *Skermanella* NA were found to be significantly more abundant when analyzed with both DESeq2 and ANCOM2. *Lactococcus lactis* and *Ligilactobacillus* sp. were observed to be more abundant in tumor samples.

### 2.4. 16S Microbiome Fresh Samples

A total of 39 fresh tissue samples were successfully processed for downstream analysis, comprising 25 normal and 14 tumor samples. The analysis approach for these fresh tissue samples was identical to that used for FFPE samples.

DNA yields from fresh tissue samples were high, but bacterial DNA yields were very low—reflected in Ct values close to those of our negative control (Table 3). Absolute taxa bar plots were generated for both phylum and species levels across all samples (Figure 2). While there was no discernible pattern based on patient ID at both levels, variations in bacterial composition or abundance between tumor and normal tissues were evident in most samples.

We further compared microbial diversity across subgroups defined by tissue type (tumor vs. normal), chemotherapy status (neoadjuvant gemcitabine-treated vs. untreated), and survival duration (long- vs. short-term survivors), but found no detectable differences in alpha- or beta-diversity metrics (Appendix A). The high p-values likely reflect microbial uniformity across these groups. However, the absence of significant differences could also be attributed to technical limitations inherent in studying low-biomass tissues such as FFPE-preserved samples, where microbial DNA fragmentation and contamination risks may affect detection sensitivity.

Differential abundance tests identified several bacterial taxa with distinct distributions across sample types and treatment subgroups (Appendix A). *Lactobacillus* sp., *Akkermansia muciniphila*, *Massilia* sp., *Corynebacterium tuberculostearicum*, and Muribaculaceae were predominantly abundant in tumor tissues, whereas *Pseudomonas* sp. and *Sporosarcina ureae* were primarily detected in normal tissues.

In patients treated with gemcitabine-based neoadjuvant chemotherapy, the relative abundance of *Pseudomonas* sp., *Corynebacterium tuberculostearicum* and *Fusobacterium nucleatum* increased, while *Duganella* sp. was more frequently detected in untreated patients. Among patients treated with gemcitabine-based chemotherapy, the short-term survival cohort was characterized by *Akkermansia muciniphila*, *Streptococcus* sp. and *Lachnospiraceae* NK4A136 group, while the long-term survival cohort showed enrichment of *Arthrobacter* sp., *Corynebacterium tuberculostearicum*, *Serratia marcescens* and *Mycobacterium* sp.

### 2.5. Gammaproteobacterial Populations

Within the FFPE samples, gammaproteobacteria were present in 15/15 tumor tissue samples, 4/5 tumor-adjacent normal tissue samples and 5/5 paraffin samples. Within tumor samples, they composed 17.5% of all bacterial sample reads.

In frozen samples, gammaproteobacteria were present in 17/17 tumor samples and 9/9 tumor-adjacent normal samples from patients treated with neoadjuvant gemcitabine-based chemotherapy. They were present in 7/8 tumor samples and 4/5 tumor-adjacent normal samples from patients who did not receive neoadjuvant chemotherapy. The mean proportion of gammaproteobacteria within tumor samples of long-term survivors treated with neoadjuvant gemcitabine-based chemotherapy was 37% vs. 44% in short-term survivors treated with neoadjuvant gemcitabine-based chemotherapy (*p* = 0.3) and 44% in patients who had not received neoadjuvant chemotherapy. Although there was significant heterogeneity in gammaproteobacterial species between patients, pseudomonal species were the most commonly identified.

## 3. Discussion

The role of the pancreatic tumor micro-organisms in promoting oncogenesis and driving response to treatment has been a topic of recent debate. Several early studies have implicated alterations in the microbiome [7,8] or mycobiome [12] in promoting pancreatic cancer oncogenesis or impacting treatment response. In particular, the pancreatic tumor microbiome has been described as distinct from surrounding tissue [11], capable of differentiating long- and short-term survivors [8] and as a potential mediator of response to gemcitabine-based chemotherapy [6,13]. Efforts to replicate these studies have been difficult [9,10], highlighting multiple limitations in the analysis of low-biomass samples. Several clinical trials are evaluating the role of the tumor microbiome in the treatment of pancreatic cancer [14,15,16,17]. Similar controversies exist in other organs. Ongoing controversy exists as to the presence of a uterine microbiome [18,19,20,21]. In this study, we found low bacterial biomass in pancreatic tumors that did not differ significantly from adjacent normal tissue, nor between long- and short-term survivors treated with gemcitabine-based neoadjuvant chemotherapy.

Our findings differ from prior preclinical studies, which suggested that intratumoral gammaproteobacteria contribute to chemotherapy resistance through cytidine deaminase activity. Specifically, Geller et al. identified gammaproteobacteria in 76% of PDAC tumors and proposed a role in metabolizing gemcitabine into its inactive form. These differences may be explained by factors such as sample preservation methods, DNA extraction protocols, and cohort characteristics [6]. Unlike our study, which included a small FFPE-based cohort, previous studies predominantly analyzed fresh frozen tissue samples, potentially capturing a broader bacterial profile.

In previous studies where DNA was extracted from FFPE tissue samples to detect the presence of bacteria, researchers often used 16S rRNA gene sequencing and qPCR methods. Although bacterial marker analysis by qPCR seemed feasible, significant challenges remained in 16S rRNA amplicon sequencing due to contamination and DNA degradation [22,23,24,25].

Given the low bacterial DNA yields observed in both FFPE and frozen samples, robust quality control measures and advanced statistical analyses were essential to ensure reliable results. We implemented stringent testing protocols, including two bacterial DNA target regions (V3V4 and V4), multiple negative and positive controls and computational contamination correction using the SCRuB pipeline [26]. To mitigate the potential impact of low DNA yields on statistical inference, we employed nonparametric tests, such as the Wilcoxon Rank Sum test for alpha-diversity and PERMANOVA for beta-diversity analysis. Differential abundance testing was conducted using DESeq2 and ANCOM-BC, with *p*-value adjustments by the Benjamini–Hochberg procedure to minimize false discoveries. Together, these measures reduced potential biases, ensuring that our findings were not artifacts of technical limitations but reflective of the true microbial composition in pancreatic tissues. In our study, DNA yield from FFPE tissue was low, and there was no significant difference in DNA yields or bacterial populations between paraffin, normal and tumor tissues. The presence of DNA yields in paraffin tissue comparable to those in tumor tissue indicates possible contamination introduced during the FFPE embedding or storage process. Low bacterial yields may be attributable to the low bacterial content of the extracted tissue or bacterial DNA degradation during storage. In frozen tissue, DNA yields were high, indicating that frozen tissue may be a better source for the study of microbial DNA. The likelihood of contamination, except for during surgical extraction, is also lower for this tissue type given that samples were immediately frozen without further processing. Ct values for bacterial DNA were more consistent for the shorter V4 region compared to the V3V4 region, highlighting the potential superiority of amplifying shorter regions of the bacterial 16S rRNA gene.

In both FFPE and frozen tissue, microbial burden was low, as indicated by high Ct counts. Microbial populations were similar in tumor and normal tissue. There was no significant difference between gammaproteobacterial population levels in long- and short-term survivors treated with neoadjuvant gemcitabine-based chemotherapy. Our inability to detect a significant correlation between gammaproteobacteria and chemotherapy response suggests that single bacterial taxa may not independently drive therapeutic outcomes. This highlights the need for broader microbiome studies integrating functional assays and multiomics profiling to capture potential host-microbiome-drug interactions. These complex dynamics underscore the importance of moving beyond single-taxon analyses toward systems-level investigations of the pancreatic tumor microenvironment. This points against, but does not exclude, gammaproteobacteria as a significant mediator of response to gemcitabine-based chemotherapy in PDAC.

Limitations of the study include the use of samples that were extracted surgically and stored in a biobank prior to use. Although samples were handled in a sterile fashion, the process of surgical extraction and storage introduced multiple opportunities for contamination. Additionally, DNA degradation likely occurred during storage, particularly in FFPE samples. Alternative DNA extraction processes, including novel approaches that separate host and bacterial cells prior to DNA extraction, may have improved data quality [27,28]. Additionally, as microbial extractions were performed from discrete locations within the tumor, they may not represent the full microbial diversity of the tumor. While nonbacterial DNA yields were high in frozen tissue, this may reflect differences in tissue composition, DNA extraction efficiency, or storage conditions. Despite standardized protocols and contamination controls, low microbial content may have been influenced by these technical or biological factors.

It is also possible that bacterial involvement in PDAC treatment response is limited to specific subpopulations of pancreatic tumors, such as advanced-stage or metastatic tumors, which were not represented in our cohort. Additionally, host factors such as immune suppression or antibiotic use may modulate tumor-associated microbial communities, complicating direct comparisons. These findings suggest the need for more granular studies involving stratified patient subgroups.

The limited sample size of this study may not have allowed for the detection of small differences between groups of interest. Regarding the evaluation of response to gemcitabine-based chemotherapy, we used surrogate outcomes such as overall and recurrence-free survival, which are influenced by multiple factors that cannot be entirely controlled. We did not evaluate tumor volumetric changes with therapy, which would be better assessed in a prospective setting.

Addressing the technical limitations associated with FFPE samples remains critical. Improving DNA and RNA extraction protocols while minimizing fragmentation and contamination will enhance the reliability of microbial profiling in low-biomass samples. Incorporating fresh frozen tissues where feasible could further improve data quality while preserving histological information. Once extraction methods are optimized, broader approaches such as metagenomic and metatranscriptomic sequencing may become feasible, enabling a deeper understanding of microbial roles in pancreatic tumors. Expanding patient cohorts with well-annotated clinical data will also strengthen investigations into host-microbiome interactions and treatment responses.

## 4. Materials and Methods

### 4.1. Patient Selection

We initially sought to optimize a protocol for 16S rRNA extraction and sequencing using formalin-fixed paraffin-embedded (FFPE) tissue while quantifying bacterial contamination within these samples. Five patients who underwent resection of PDAC were selected, and FFPE blocks were reviewed by an attending pathologist to confirm tumor tissue, tumor-adjacent normal pancreas and areas of paraffin without tissue from blocks containing tumor tissue. For each of these 5 patients, we performed DNA extractions and performed 16S rRNA sequencing, as described below, on three areas of tumor tissue, two areas of normal tissue and one area of adjacent paraffin (Table 2).

Given the possibility of bacterial contamination during the fixation and paraffin embedding process, we proceeded with subsequent studies using frozen tissues. As these samples were immediately frozen after collection and were confirmed to have no subsequent defrosting-refreezing events, such samples should allow for minimal contamination and maximal preservation of RNA within tissue samples. All samples were primary PDAC resections. We reviewed patients with resected pancreatic cancer for demographic, clinical, surgical and survival variables [29]. We selected 18 patients treated with neoadjuvant gemcitabine-based chemotherapy regimens in which 9 of these patients were long-term survivors (LTS; 31–83 months from surgery to last follow up) and 9 were short-term survivors (STS; 5–17 months from surgery to death). We additionally selected 8 control patients who had not received neoadjuvant chemotherapy (Table 1). For 5 patients from each group (LTS, STS, untreated), tumor and normal samples underwent 16S rRNA extraction and sequencing. For the remainder of the samples, only tumor samples were included. All patient review was approved under Columbia University IRB AAAU0134. mOS = median overall survival from time of diagnosis, mRFS = median-recurrence free survival from time of diagnosis, GTX = gemcitabine, docetaxel and capecitabine, FFPE = formalin fixed and paraffin embedded, LTS = long-term survivors treated with gemcitabine-based neoadjuvant chemotherapy, STS = short-term survivors treated with gemcitabine-based neoadjuvant chemotherapy.

### 4.2. Sample Collection

Sample collection from identified patients was approved in IRB AAAT6521. All samples were collected in the operating room during resection of pancreatic tumors, using sterile procedures. FFPE samples were immediately preserved in paraffin blocks and stored in Columbia’s tumor biorepository at room temperature. Frozen samples were extracted using sterile operating room procedures and frozen immediately after collection. No samples were defrosted prior to processing for this study. All sample processing was performed manually using sterile latex gloves.

Due to inherent differences between FFPE and fresh frozen tissues, including DNA fragmentation, chemical cross-linking, and varying extraction efficiency, microbial DNA yields from FFPE samples were expected to be lower. To account for this, we performed separate analyses for FFPE and fresh frozen samples, including distinct sequencing runs and normalization steps. Additionally, we targeted two 16S rRNA gene regions (V3-V4 and V4) to accommodate DNA fragmentation in FFPE samples. This helped ensure reliable amplification while controlling for technical biases.

### 4.3. DNA Isolation

A total of 69 tissue samples were used in this study, comprising 30 FFPE tissue samples and 39 fresh frozen tissue samples. Genomic DNA was extracted using the QIAamp^®^ 96 DNA QIAcube^®^ HT (Qiagen, Germany) following the manufacturer’s guidelines.

For FFPE tissues, a specialized pretreatment was employed to optimize DNA purification from these fixed samples. Each FFPE sample (≤25 mg) was washed twice with phosphate-buffered saline (PBS) to remove residual fixatives. After discarding the PBS, tissue digestion was performed using proteinase K and ATL buffer under heated conditions, followed by DNA purification per the manufacturer’s recommended protocol. Fresh tissue samples underwent DNA extraction directly without the FFPE pretreatment, adhering strictly to the manufacturer’s instructions. Both frozen and FFPE extractions included a positive control ZymoBIOMICS Microbial Community Standard, D6300 (ZymoBIOMICS, Irvine, CA, USA) and two negative controls (HyClone™ Molecular Biology Grade Water, Cytiva, Marlborough, MA, USA). DNA was eluted in 100 µL of RNase/DNase-free buffer, and DNA yield was quantified using a Qubit 2.0 fluorometer (Invitrogen, Carlsbad, CA, USA).

### 4.4. 16S rRNA Gene Amplification

Owing to the inherent DNA fragmentation observed in FFPE samples, where the extracted DNA typically remains under 650 bp in length, there’s a potential inadequacy in using V3-V4 primers to span the intended target region (341f-806r/606bp). To address this and ensure comprehensive coverage, we also targeted the 16S V4 region (525f-806r/388bp) for amplification. We employed both the Zymo Quick-16S/ V3V4 Plus NGS Library prep kit and the Quick-16STM Plus NGS Library Prep Kit (V4) from Zymo Research (ZymoBIOMICS, Irvine, CA, USA), aiming to optimize bacterial 16S rRNA gene amplification and ascertain the most suitable region for this study. The chosen library preparation kits incorporated the equalase technique, offering a dual advantage: amplifying the 16S rRNA gene with precision using qPCR while also facilitating its subsequent indexing.

PCR amplifications for both target regions were conducted using the CFX Opus 96 Real-Time PCR System (Bio-Rad, Hercules, CA, USA). Postamplification, the resultant 16S libraries were pooled by adding equal volume, and the final library concentration was quantified utilizing the Qubit 2.0 fluorometer (Invitrogen). After equimolar pooling based on these determinations, sequencing was carried out on the Illumina MiSeq platform at an 8pM loading concentration supplemented with 20% PhiX, using the paired-end 300 cycle MiSeq Reagent Kit V3 (Illumina, San Diego, CA, USA).

### 4.5. 16S rRNA Analysis

16S rRNA sequences from both targets (V3-V4 and V4) were processed and applied using the DADA2 (DADA2 1.12.1) pipeline [30] in conjunction with R v4.1.0. DADA2 was employed for quality filtering, trimming, error correction, exact sequence inference, chimera removal and generation of the amplicon sequence variant table (ASV) with a minimum count cutoff set at 7500. Based on the quality score profiles of the sequencing reads, forward and reverse reads were truncated at 240 bp prior to merging. Ambiguities in the overlap region were not permitted, and default parameters were utilized in the R DADA2 package. After the dereplication and merging of reads, chimeric reads were identified by consensus across samples using the DADA2 function removeBimeraDenovo. All samples passed the set threshold of 5000 reads post-quality filtering for inclusion in the analysis.

After DADA2 pipeline processing, any mitochondrial reads were identified and removed to ensure only microbial genomic data was assessed. To minimize contamination effects, we applied the SCRuB [26] contamination modeling pipeline. SCRuB distinguishes genuine bacterial DNA from environmental contaminants by leveraging microbial composition patterns in negative controls and sample-specific read counts. We included extraction-negative and PCR-negative controls in every sequencing batch to create a background contamination profile. SCRuB iteratively filters contaminants by adjusting for microbial taxa that are over-represented in control samples, ensuring that only biologically relevant taxa are retained for downstream analysis.

The MAFFT and FastTree modules in QIIME2 [31] facilitated the generation of a phylogenetic tree of all ASV sequences. Taxonomic classification was undertaken using a native naïve RDP Bayesian classifier aligned against the Silva version 138 database [32].

### 4.6. Statistical Analysis

Survival analyses were calculated using Kaplan–Meyer statistics. Descriptive statistics were utilized to summarize the demographic information for the entire cohort. This data was integrated into R (version 4.1.0) to carry out analyses on α-diversity metrics, such as Shannon and Chao, and Principal Coordinates Analysis (PCoA) based on the unifrac distance matrix. These analyses were conducted using the phyloseq package (v1.36.0) in R [33].

Sample size was determined based on the availability of PDAC samples from our biorepository and the feasibility of performing microbial DNA extraction and sequencing. While we acknowledge the limited sample size, prior microbiome studies have used similar cohort sizes due to technical constraints inherent in working with FFPE and frozen tissue samples. To maximize statistical power, we applied stringent filtering criteria, including PERMANOVA for beta-diversity analysis, Wilcoxon Rank Sum tests for alpha-diversity, and DESeq2 [34] and ANCOM-BC [35] for differential abundance testing with adjusted p-values using the Benjamini-Hochberg method.

## 5. Conclusions

We found low microbial content of tumor and normal tissue in resected PDAC samples using multiple sample types and extraction techniques. In frozen tissue, nonbacterial DNA yields were high, indicating that this low content was not a failure of our extraction techniques. In FFPE, microbial content of tumor-adjacent paraffin had similar absolute abundance to tumor tissue, indicating that a proportion of the documented pancreatic tumor microbiome may be due to contamination. Together, our findings do not indicate a unique tumor-associated pancreatic cancer microbiome, nor one that appears to play a significant role in response to gemcitabine-based chemotherapy.

## Figures and Tables

**Figure 1 ijms-26-00140-f001:**
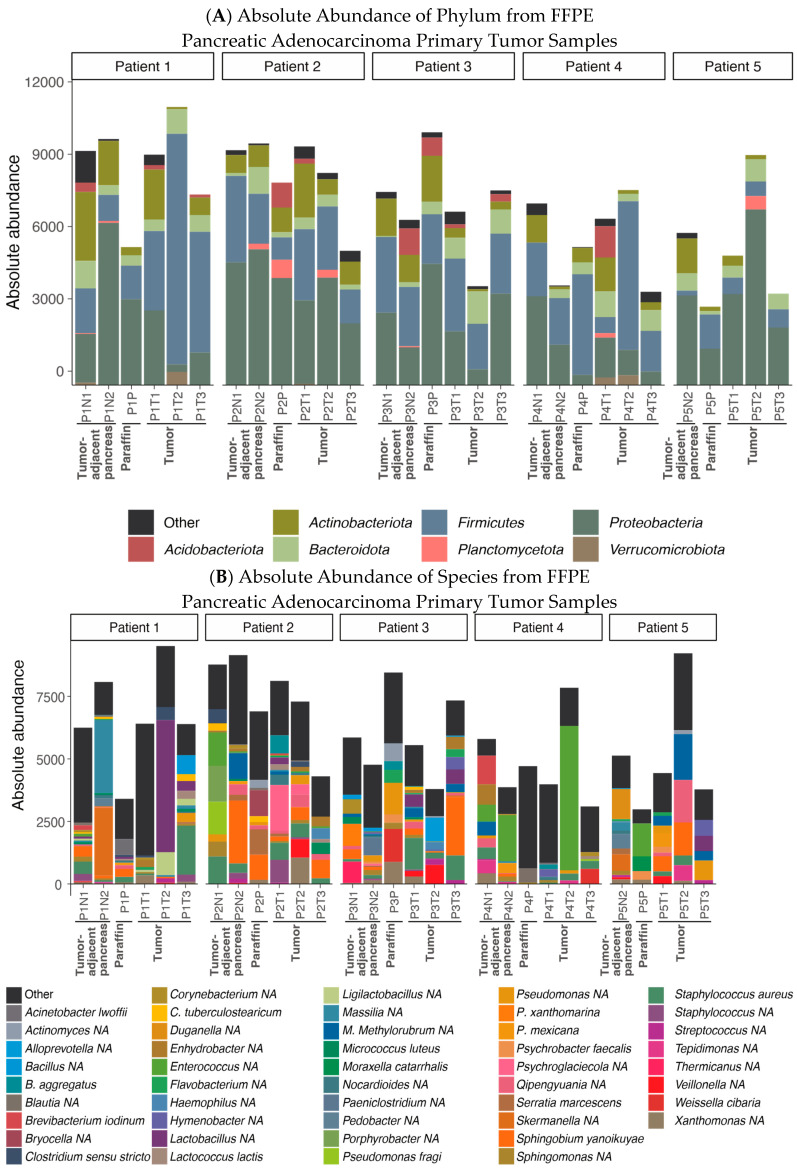
Absolute taxa bar plot in FFPE samples. (**A**) Phylum level representation. (**B**) Species level representation.

**Figure 2 ijms-26-00140-f002:**
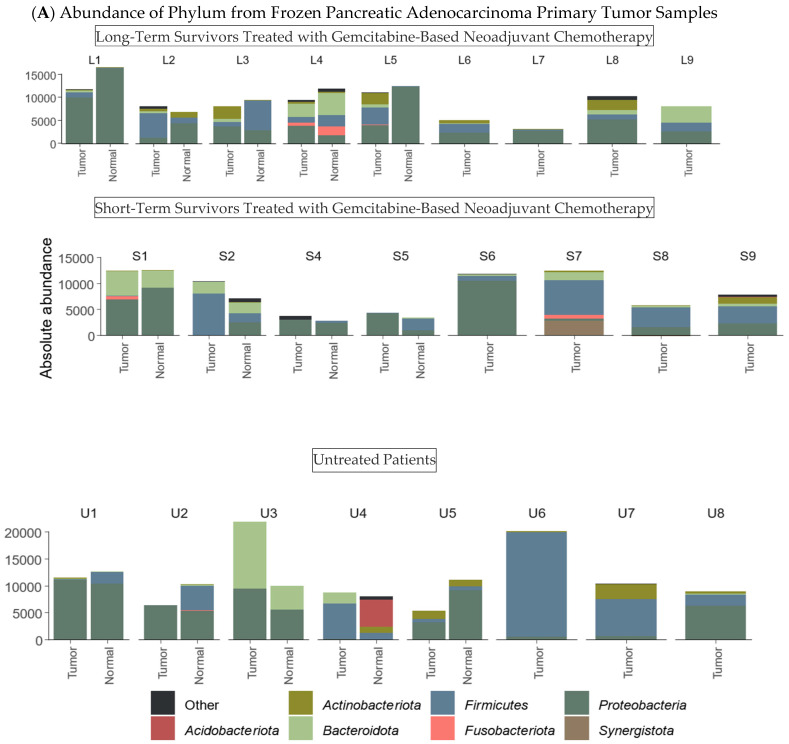
Absolute taxa bar plot in the fresh frozen samples. (**A**) Phylum level representation. (**B**) Species level representation.

**Table 1 ijms-26-00140-t001:** Patient characteristics.

	Median Age at Diagnosis	Neoadjuvant Chemotherapy		Head of Pancreas	Adjuvant Chemotherapy	mOS (Years)	mRFS (Years)
		GTX	Gemcitabine Nab-Paclitaxel				
FFPE Samples (*n* = 5)	67	0	3 (60%)	4 (80%)	3 (60%)	4.0	1.3
Frozen Samples (*n* = 25)							
LTS (*n* = 9)	68	6 (67%)	3 (33%)	5 (56%)	4 (44%)	6.6	6.6
STS (*n* = 8)	71	4 (50%)	4 (50%)	5 (63%)	4 (50%)	1.4	1.1
Untreated (*n* = 8)	67	0	0	7 (88%)	7 (88%)	4.0	1.3

**Table 2 ijms-26-00140-t002:** FFPE sample qPCR data shows the Ct values and starting quantity for each sample.

		DNA Yield (ng/µL)	Bacterial qPCR Ct		16S V4 Sequencing Unique Reads
			V3V4	V4	
Controls	Negative	0.1	NA	38.3	13,489
	Positive	3.12	18.1	17.1	11,329
Patient 1 Normal Tissue	P1N1	1.0	NA	35.5	11,329
	P1N2	0.9	NA	34.3	11,342
Patient 1 Paraffin	P1P	1.7	NA	37.7	12,254
Patient 1 Tumor Tissue	P1T1	4.4	NA	36.8	12,730
	P1T2	0.3	NA	33.5	12,335
	P1T3	NA	NA	36.3	11,495
Patient 2 Normal Tissue	P2N1	0.7	NA	37.2	15,311
	P2N2	0.1	NA	36.7	14,104
Patient 2 Paraffin	P2P	0.4	NA	37.4	11,579
Patient 2 Tumor Tissue	P2T1	1.1	NA	35.7	11,471
	P2T2	1.1	NA	36.4	12,289
	P2T3	NA	NA	37.1	15,228
Patient 3 Normal Tissue	P3N1	NA	NA	37.6	13,311
	P3N2	1.7	NA	34.3	11,373
Patient 3 Paraffin	P3P	3.4	NA	39	15,519
Patient 3 Tumor Tissue	P3T1	6.2	NA	36	12,938
	P3T2	9.8	NA	31.6	11,553
	P3T3	NA	NA	37.7	10,984
Patient 4 Normal Tissue	P4N1	0.2	NA	38.2	14,623
	P4N2	0.4	NA	37.6	15,922
Patient 4 Paraffin	P4P	0.3	NA	36.6	12,010
Patient 4 Tumor Tissue	P4T1	5.3	NA	NA	7908
	P4T2	6.4	NA	36	11,123
	P4T3	NA	NA	NA	10,049
Patient 5 Normal Tissue	P5N1	0.7	NA	41.8	8972
	P5N2	0.1	NA	36.6	13,041
Patient 5 Paraffin	P5P	2.2	NA	37.1	14,720
Patient 5 Tumor Tissue	P5T1	0.2	NA	38.7	9781
	P5T2	2.6	NA	38.3	16,077
	P5T3	NA	NA	39	13,630

**Table 3 ijms-26-00140-t003:** Frozen sample qPCR data shows the Ct values and starting quantity for each sample.

		DNA Yield (ng/µL)	Bacterial qPCR		16S V4 Sequencing Unique Reads
			V3V4 Ct	V4 Ct	
Controls	Negative	0.1	39.7	37.6	13,489
	Positive	11.3	17.1	16.4	11,992
Long-Term Survivors, Tumor-Adjacent Normal Tissue	LN1	66.6	36.2	33.7	17,042
	LN2	4.7	40.5	39.9	15,425
	LN3	33.1	37.8	36.6	10,333
	LN4	11.5	30.4	28.3	14,459
	LN5	18.4	38.1	38.7	14,195
Long-Term Survivors, Tumor Tissue	LT1	15.9	38.4	36.1	13,030
	LT2	12	40.2	38.4	13,483
	LT3	47.7	39.3	39.3	12,572
	LT4	25.4	38.6	38.3	13,310
	LT5	9.3	40.1	35.4	12,300
	LT6	33.7	39.5	39.2	11,798
	LT7	8	41.3	40.2	10,804
	LT8	38.7	40.5	38.4	12,471
	LT9	11.2	41.4	39.9	16,723
Short-Term Survivors, Tumor-Adjacent Normal Tissue	SN1	8.1	36.9	37.4	13,527
	SN2	73	n/a	38.5	13,753
	SN4	10.6	40.8	40.1	13,182
	SN5	12.3	39.1	39.4	14,197
Short-Term Survivors, Tumor Tissue	ST1	7.6	37.7	36.7	13,932
	ST2	10.5	40.9	31	11,158
	ST4	20.9	n/a	40	11,403
	ST5	44.8	39.3	38.8	14,897
	ST6	8.8	37.3	35.3	15,870
	ST7	14.3	32.6	31.6	13,685
	ST8	56.2	36.4	34.7	11,210
	ST9	7.3	40.8	35.7	9641
Patients with no Neoadjuvant Treatment, Tumor-Adjacent Normal Tissue	UN1	33.8	37.3	34.4	16,174
	UN2	38.8	39.1	36.5	11,703
	UN3	46.4	34	33.1	10,044
	UN4	98	40.8	37.9	11,709
	UN5	15.4	40	38.5	15,618
Patients with no Neoadjuvant Treatment, Tumor Tissue	UT1	48.7	38.5	35.3	13,575
	UT2	21.9	40.9	39.5	18,306
	UT3	31	34.5	33.6	22,619
	UT4	51.9	36.4	32.1	9514
	UT5	33.4	37.5	42.1	7823
	UT6	15	37.1	36.1	23,662
	UT7	0.1	39.8	40.1	14,770
	UT8	47.2	39.2	39.7	10,442

## Data Availability

The data presented in this study are available on request from the corresponding author due to the presence of patient health information requiring deidentification.

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
