# Peer review of "Low Bacterial Biomass in Human Pancreatic Cancer and Adjacent Normal Tissue"

_ijms, 2024, doi:10.3390/ijms26010140_

Round 1

Reviewer 1 Report

Comments and Suggestions for Authors

The article discusses the role of the microbiome in pancreatic cancer, particularly focusing on the impact of the microbiome on chemotherapy response. The study found a low microbial load in pancreatic tumor tissue, which may be partially attributable to contamination. The findings interesting, and it do not support a significant role of the microbiome in mediating response to gemcitabine-based chemotherapy. Please attention some issues.

1. The study used samples from human pancreatic tumor resections and performed 16SrRNA sequencing on formalin-fixed paraffin-embedded (FFPE) and fresh frozen human PDAC resection samples.  The number of FFPE is 5. Whether the sample too little?

2. The bacterial DNA yields were generally low, comparable to those seen in negative controls. So the the testing method, sample quality control, and statistics are crucial. Please emphasize these aspects in the discussion.

3. The study also found that gammaproteobacteria concentrations did not correlate with outcomes in patients treated with neoadjuvant gemcitabine-based chemotherapy. But many preclinical studies have implicated the pancreatic tumor microbiome driving response to therapy. How to view and evaluate the future of intratumoral microbial therapy? 

Reviewer 2 Report

Comments and Suggestions for Authors

This manuscript investigates the bacterial biomass within human pancreatic cancer and adjacent normal tissues using 16S rRNA sequencing in both formalin-fixed paraffin-embedded (FFPE) and fresh frozen samples. The study reports low bacterial biomass in these tissues, suggesting that microbiome abundance in pancreatic cancer may not significantly influence response to gemcitabine-based chemotherapy. These findings challenge previous hypotheses about the role of intratumoral bacteria in pancreatic cancer progression and treatment response.

Abstract

The abstract is descriptive but lacks precision. Instead of stating, "We performed 16S rRNA sequencing," it would be clearer to specify the exact methodologies used, such as "16S rRNA gene amplification and sequencing on both FFPE and fresh frozen samples." Additionally, the conclusion should explicitly state the clinical implications of the findings, particularly their relevance to current hypotheses about the microbiome's role in chemotherapy response. For example, it should state that no correlation was found between bacterial biomass and chemotherapy outcomes, thus challenging prior assumptions.

Introduction

The introduction discusses the role of microbiomes in carcinogenesis but fails to adequately connect prior research to the hypothesis tested. The hypothesis—that bacterial populations, particularly gammaproteobacteria, influence chemotherapy outcomes—is presented but requires deeper justification. Including more detailed references to earlier preclinical studies and their limitations would provide a better foundation. For instance, mention how contamination issues in microbiome studies have skewed interpretations and how this study seeks to address them.

Materials and Methods

This section requires significant expansion for reproducibility. The methodology for distinguishing contamination from genuine bacterial DNA is not sufficiently robust. The application of the SCRuB technique should be better explained, including how it ensures the accuracy of sequencing results. Additionally, details about the statistical power and how sample size was determined should be included, as the limited sample size raises concerns about the robustness of the findings. Another key issue is the lack of a discussion on potential biases introduced by using FFPE and fresh frozen samples, which may yield differing DNA quality and quantity.

Results

The results section does not sufficiently elaborate on the statistical analysis or its limitations. For example, the non-significant p-values for diversity indices should be critically interpreted. Why were these measures chosen, and what do the high p-values imply about microbial uniformity across tissue types? Additionally, the presentation of bacterial taxa abundance is not sufficiently detailed. Tables and figures lack clarity; for instance, taxa like "Akkermansia muciniphila" and "Pseudomonas sp." should be linked more explicitly to specific cohorts to allow the reader to discern any trends.

Discussion

The discussion underplays the implications of low bacterial biomass in pancreatic cancer tissue. The manuscript should delve deeper into how these findings compare with previous studies that identified gammaproteobacteria as contributors to chemotherapy resistance. It should also consider alternative explanations for the results, such as the possibility that bacteria play a more significant role in subpopulations of pancreatic tumours not captured in this study. Furthermore, the authors should propose concrete next steps for research, such as improving sample preservation techniques or employing metagenomic approaches for more comprehensive analyses.

Each query raised must be addressed comprehensively, and all changes highlighted in yellow within the revised manuscript. 

Reviewer 3 Report

Comments and Suggestions for Authors

In manuscript ID: “ijms-3366202” the authors investigate the role of the microbiome, particularly intratumoral gammaproteobacteria, in mediating response to gemcitabine-based chemotherapy in pancreatic ductal adenocarcinoma (PDAC). Although the study is well-conducted, addressing the following issues before considering publication would be beneficial.

1.    Please include a detailed description of the key steps in the method section “2.3. DNA isolation”.

2.    Section “2.4. 16S rRNA Gene Amplification”, As the V3-V4 and V4 primer sets target different regions of the 16S rRNA gene, could the authors clarify the criteria used to assess the adequacy of amplification for each target region?

3.    Could the proportion of gammaproteobacteria be a biomarker for treatment response or prognosis?

4.    Are there specific species or strains of gammaproteobacteria associated with tumor tissues, and what is their functional role?

5.    Given the lack of significant difference in gammaproteobacterial population levels between long- and short-term survivors treated with gemcitabine, could other microbial populations or host factors play a more prominent role in the response to chemotherapy?

6.    Given that non-bacterial DNA yields were high in frozen tissue, how can we rule out other factors (e.g., tissue type, storage conditions) that may contribute to the low microbial content observed?

Round 2

Reviewer 2 Report

Comments and Suggestions for Authors

Thank you for your revisions which I accept